# Super-Resolution Microscopy to Study Interorganelle Contact Sites

**DOI:** 10.3390/ijms232315354

**Published:** 2022-12-05

**Authors:** Jon Ander Nieto-Garai, June Olazar-Intxausti, Itxaso Anso, Maier Lorizate, Oihana Terrones, Francesc-Xabier Contreras

**Affiliations:** 1Department of Biochemistry and Molecular Biology, Faculty of Science and Technology, University of the Basque Country (UPV/EHU), Barrio Sarriena s/n, 48940 Leioa, Spain; 2Structural Glycobiology Laboratory, Biocruces Bizkaia Health Research Institute, Cruces University Hospital, 48903 Barakaldo, Spain; 3Instituto Biofisika (UPV/EHU, CSIC), Barrio Sarriena s/n, 48940 Leioa, Spain; 4Ikerbasque, Basque Foundation of Science, 48011 Bilbao, Spain

**Keywords:** super-resolution microscopy, SIM, TIRFM, STED, SMLM, PALM, STORM, organelles, membrane contact sites

## Abstract

Interorganelle membrane contact sites (MCS) are areas of close vicinity between the membranes of two organelles that are maintained by protein tethers. Recently, a significant research effort has been made to study MCS, as they are implicated in a wide range of biological functions, such as organelle biogenesis and division, apoptosis, autophagy, and ion and phospholipid homeostasis. Their composition, characteristics, and dynamics can be studied by different techniques, but in recent years super-resolution fluorescence microscopy (SRFM) has emerged as a powerful tool for studying MCS. In this review, we first explore the main characteristics and biological functions of MCS and summarize the different approaches for studying them. Then, we center on SRFM techniques that have been used to study MCS. For each of the approaches, we summarize their working principle, discuss their advantages and limitations, and explore the main discoveries they have uncovered in the field of MCS.

## 1. Introduction

The well-defined images of isolated organelles that many textbooks still display today have been rendered obsolete. The development of high-resolution imaging techniques has allowed direct visualization of intracellular structures, and has uncovered a highly dynamic and complex membranous network formed by interacting organelles. Although the vicinity between organelles was first described in the late 1950s [1,2], no function was assigned to the contacting areas until almost 40 years later when it was discovered that the synthesis and transport of diverse phospholipids occur at the juxtaposition between the Endoplasmic Reticulum (ER) and mitochondria [3]. This finding flipped the research interest from identifying each compartment’s distinguishing feature to the functional study of the interorganelle membrane contact sites (MCS).

MCS are defined as areas of close proximity (from 10 to 80 nm) between the membranes of two organelles [4]. The structural maintenance of these sites requires molecular bridges that are referred to as tethers. Tether proteins are responsible for physically bridging interacting membranes while maintaining each organelle´s structural integrity, thus avoiding membrane fusion [5]. In addition to these scaffold proteins, MCS have a unique protein composition that serves as a functional hub for many physiological processes. Functions of MCS-residing proteins can be grouped into (i) bidirectional transport of molecules, such as Ca^2+^ or phospholipids, and (ii) transmission of environmental signals, including those that regulate organelle morphology and biogenesis [6]. As an example of the latter, it is well established that ER-constricted mitochondrial sites serve as a regulatory center for mitochondrial fission [7]. At these sites, actin-nucleating proteins promote actin assembly to generate small patches of actin–myosin II cytoskeleton that are required for mitochondrial pre-constriction and subsequent recruitment of the fission executor protein Drp1 [8,9,10]. In essence, MCS can be seen as nanometer-scale structures responsible for functionally coupling contacting organelles.

While proximity-dependent biotinylation methods (APEX2, BioID) have unveiled the proteome of some MCS [11,12,13,14,15], the lipidome of most of them is still unknown. ER membranes contacting mitochondria have the characteristics of an intracellular lipid raft, a membrane domain rich in cholesterol and sphingolipids with a liquid-ordered structure [16,17,18,19,20,21]. This lipid composition that rigidifies the membrane could be required to generate or stabilize MCS. However, whether this lipid composition is a feature shared by all MCS remains to be clarified.

Several years after identifying the ER-mitochondria MCS as regulatory centers of phospholipid homeostasis, these regions were also found to be essential for Ca^2+^ transport between the organelles [22]. Since these first observations, these sites’ functional and molecular characterization has gained much interest, to the extent that some authors have even proposed a new research field termed *contactology* [23]. To date, the best-characterized MCS are those involving the ER, which we now know, in addition to mitochondria, also interacts with the plasma membrane (PM), Golgi apparatus, peroxisomes, lysosomes, lipid droplets (LD), and endosomes [24]. These close appositions with ER are essential for many cellular functions. For instance, ER-mitochondria MCS regulate, in addition to the aforementioned phospholipid and Ca^2+^ homeostasis, cell death, mitochondrial dynamics and biogenesis, and autophagy [23]. This latter function is also regulated by ER-PM juxtapositions [25]. Many other MCS that do not implicate ER have been discovered, and their roles in cellular fate uncovered, evidencing that virtually all organelles are physically tethered to each other [26].

As MCS fulfill relevant cellular functions, their regulation is crucial, and hence, dysregulation leads to cellular impairment. MCS dysfunction has been related to the development of different pathologies such as diabetes [27], cancer [28], and neurodegenerative disorders [29,30], among others. Given the importance of MCS for cellular homeostasis and the potential relation to multiple pathogeneses, the need for reliable knowledge of the molecular mechanism governing MCS function, formation, and stability seems indisputable. In this regard, an enormous effort has recently been made to generate new approaches or adapt existing ones to study these nanoscopic regions specifically [26].

The first evidence pointing to an interaction between two organelles (ER and mitochondria) was obtained thanks to electron microscopy (EM) [1,2]. Although essential for the finding and routinely used to study MCS, EM requires chemical fixation that, on the one hand, hampers dynamic studies and, on the other, may affect interorganelle apposition and thus result in an underestimation of the number of MCS. Nowadays, variants of EM, such as electron tomography (ET) or cryo-ET, have allowed high-resolution and label-free three-dimensional reconstruction of ER-PM and ER-mitochondria MCS [31,32,33]. Shortly after ER-mitochondria contacts were discovered, they were biochemically characterized. ER-mitochondria characterization required a previous isolation step by cellular fractionation and density gradient centrifugation [34]. This purification step generates contaminants and loss of key components, and has to be designed considering that isolated MCS have features of both organelles. Other approaches, such as proximity labelling techniques, have overcome the requirement of isolating the compartment of interest. Methods based on the functional complementation of the enzymatic activity of APEX2 peroxidase or BirA biotin ligase have allowed successful proteomic characterization of the ER-mitochondria and ER-peroxisome MCS [11,12,13,14,15]. In these approaches, an enzyme fragment is targeted to each interacting membrane allowing functional complementation exclusively at the MCS, where the two fragments are in close proximity. In both approaches, the enzymatic activity leads to specific biotinylation of surrounding proteins that can be easily extracted using well-established streptavidin systems.

Proximity-driven signal generation assays, in combination with confocal microscopy, have also been used to visualize MCS. For example, fluorescence resonance energy transfer (FRET), proximity ligation assay (PLA), bimolecular fluorescence complementation (BiC), and dimerization-dependent fluorescent protein (ddFP) assays have been widely used to extract quantitative data concerning the extent of the contacting area and the distance between organelles [26]. Although these methodologies, based on the detection of fluorescence emission, are fast, easy to handle, and can be applied to living cell measurements, signal detection by confocal microscopy entails an inherent problem: most MCS are far below the optical diffraction limit. Some recent advances have allowed modest improvements in the lateral resolution of confocal set-ups and have been specifically applied to study MCS. Cook et al. extensively characterized the restructuration of several MCS during cytomegalovirus infection. The virus hijacks mitochondria-ER MCS to form specific structures, known as mitochondria-ER encapsulation structures (MECSs), to support the production of new viral particles. Cytomegalovirus also influences ER-peroxisome interactions to induce peroxisome growth and allow for a more efficient virion assembly [35]. Nevertheless, most confocal set-ups remain limited for MCS studies, and other imaging techniques must be used. To answer this challenge, super-resolution microscopy techniques were developed to break the 200 nm resolution limit established by diffraction. These techniques that provide a sub-diffraction-limit spatial resolution (20 to 100 nm) have shed some light on the structural organization and dynamics of MCS (Figure 1).

In this review, we first summarize the different super-resolution techniques employed to study interorganelle interactions, and their methodological bases, advantages, and limitations are discussed. Then, the information about MCS obtained with each of them is mentioned.

## 2. Super-Resolution Fluorescence Microscopy to Study MCS

SRFM comprises a series of techniques that allow obtaining images with spatial resolutions beyond those provided by conventional optical microscopy. In conventional optical microscopy, the optical resolution—the minimal distance at which two individual objects can be differentiated—is limited by the diffraction of light. When light travels through the microscope, it interacts with the lens and is diffracted in a pattern called Airy disks, a bright region in the center surrounded by a series of concentric rings of decreasing intensity. In a fluorescence microscope, this diffraction affects both the excitation and the emission processes. During the excitation process, a laser beam is focused through the objective to a focal point within the sample, but the objective diffracts this beam and widens it to a minimal excitation spot size of 460–850 nm for visible light. During the emission process, Airy disks created by diffraction blur out any light emitted by a single fluorophore to a certain minimal size bigger than the emitter—commonly referred to as the Point Spread Function (PSF) (Figure 2a) [36]. Because of this diffraction, if the distance between two fluorophores is too small, their PSFs overlap, and the fluorophores are seen as a single object in the resulting image (Figure 2b). The minimal theoretical distance at which two objects can be resolved is called the diffraction limit, and it depends on the wavelength of the light and the properties of the objective lens. In laser scanning confocal microscopy (LSCM), it has values of >200 nm in the lateral plane and >500 nm in the axial direction [37]. Conventional optical microscopy techniques cannot differentiate any two molecules closer to this distance.

Because the average distance between organelles is 10–80 nm [4], and the tethers populating the MCS are even smaller, conventional optical microscopy techniques are not suitable for the in-depth characterization of MCS (Figure 2c). To visualize these nanometer-scale structures with resolutions of up to ≈20 nm, SRFM approaches are required. These techniques employ a variety of approaches to circumvent the limit established by the diffraction of light and involve modifications in the excitation and emission of the fluorescent molecules. Of the different available techniques, MCS have been primarily studied by four main methodological approaches: Structured Illumination Microscopy (SIM); Total Internal Reflection Fluorescence Microscopy (TIRFM); Stimulated Emission Depletion (STED) microscopy; and the related Single Molecule Localization Microscopy (SMLM) techniques that include Photo-Activated Localization Microscopy (PALM), STochastic Optical Reconstruction Microscopy (STORM) and Points Accumulation for Imaging in Nanoscale Topography (PAINT). The main characteristics, advantages, and limitations of these techniques when applied to the study of MCS are summarized in Table 1.

### 2.1. SIM

SIM is an interferometric technique that extracts super-resolution information about the structures in the sample from low-resolution interference patterns. In fluorescence microscopy, at any given point, the intensity of the fluorescence depends on both the intensity of the excitation light and the density of the probe at that point. In SIM, the sample is excited by a high-frequency grid-like patterned widefield illumination (i.e., an excitation light with grid-shaped alternating high- and low-intensity bands), which, combined with variations in fluorescence caused by the structures in the sample, creates interference patterns that can be used to extract information about the probe density. By rotating the patterned excitation light, several interference patterns can be obtained and combined to reconstruct a super-resolved image with up to ≈100 nm lateral resolution (Figure 3) [62]. SIM does not require specialized fluorophores or labelling techniques, has high sensitivity and contrast, and can be combined with various complementary approaches to extract additional information. For example, this technique can be used to reconstruct 3D images (3D-SIM), which doubles the axial resolution to ≈300 nm and provides further information about biological structures [63].

When applied to MCS, the main limitation of SIM is that it only provides ≈100 nm lateral resolution, which limits its use to only the largest MCS structures or general morphological studies, although this resolution can be further improved up to ≈50 nm with an approach known as non-linear or Saturated SIM (SSIM) [64]. Additionally, when studying dynamic MCS processes, it must be considered that the quality of the images depends on the number of illumination patterns applied in each frame, which in turn increases acquisition time, meaning that a trade-off exists between spatial and temporal resolutions.

Even with the limitations mentioned above, since this technique only requires a simple widefield microscope, can be used with only a single laser, requires no specialized fluorophores, and, in the right conditions, can provide decent temporal resolution, it has been widely applied in the study of MCS in both fixed and living cells. By combining 3D-SIM and Grazing Incidence—a technique similar to the TIRFM technique detailed below—SIM (GI-SIM), ER areas previously thought to be flat sheet membranes were discovered to consist of highly dynamic tubular structures [38]. Although the authors did not delve into their function, they postulated that ER tubules could allow the storage of excess membrane proteins and lipids necessary for modulating interactions with other organelles such as mitochondria or endosomes. Later, the application of GI-SIM confirmed that these tubular ER structures regulate mitochondria fission and fusion through ER-mitochondria MCS [39]. This study also revealed that ER-organelle contacts allow the ER to indirectly use molecular motors in a hitchhiking process, in which the ER uses MCS to follow the movement of contacting organelles to rearrange and reshape its structure. ER-mitochondria MCS are also actively involved in the replication and distribution of nucleoids, chromosome-like organellar nuclei that contain the mitochondrial DNA (mtDNA). Segregation of nucleoids has traditionally been thought to be passively determined by mitochondrial fusion and fission, but the application of GI-SIM has recently determined that ER-mitochondria MCS serve as a platform for the active transport of nucleoids through mitochondrial tubulation processes [40]. Additional applications of SIM and 3D-SIM have also established that ER-mitochondria MCS are involved in other biological functions and have identified several specific tethers, such as Stasimon [41] and Mmr1 [42]. The application of 3D-SIM has described that the ER protein protrudin mediates ER-endosome MCS, allowing the transport of late endosomes to the PM, promoting protrusion and neurite outgrowth [44].

The ER is an important player in regulating protein folding and secretion, which requires an appropriate ATP supply. Experiments using dual-color SIM have demonstrated that ER stress induced by the accumulation of unfolded glycoproteins increases the stability and the lifetime of contacts between mitochondria and ER [65]. This physical and temporal enhancement in the connectivity is driven by the ER and mitochondrial-residing tether protein Mfn2 and leads to an increase in (i) basal mitochondrial Ca^2+^ levels, (ii) mitochondrial metabolisms, and (iii) ATP production. Since hydrolysis of ATP is required for both ER chaperons-driven protein folding and protein trafficking, the unfolded protein response also increases ATP trafficking towards the ER in tunicamycin-treated HeLa cells [65].

SIM has also been applied for studying other MCS. In a dual-color SIM approach, authors designed fluorophores targeted explicitly to mitochondria or lysosomes and imaged lysosome–mitochondria MCS for long periods of time (up to 13 min) without significant photobleaching. This approach identified four types of physical interactions between lysosomes and mitochondria, ranging from superficial adhesion between organelles to a more intimate entrapment of lysosomes in the mitochondrial network [43]. It has recently been established that these interconnectivities regulate molecular transport between the organelles and lysosomal dynamics and morphology. Generating a dichromatic probe that fluoresces at different wavelengths depending on localization (due to the effect of pH in probe reactivity) and binding to HSO_3_^−^, Fang G. et al. visualized and monitored the transport from mitochondria to lysosomes of reactive sulfur species (RSS) that are generated by the metabolisms of thiol-containing amino acids [66]. In addition, SIM has allowed it to be established that mitochondria–lysosome MCS regulate lysosomal network dynamics through the mitochondrial fission proteins Fis1 and Mid51 [67].

Other discoveries using SIM include the identification of the Rab18-NRZ-SNARE complex as one of the tethers involved in ER-LD MCS that regulates LD growth [45], and that VAP-ACBD5-mediated ER-peroxisomes MCS regulate peroxisome growth, the synthesis of plasmalogen phospholipids, and the maintenance of cellular cholesterol levels [13].

Recently, SIM has been combined with optical diffraction tomography (ODT) techniques in an approach termed Super-Resolution Fluorescence-Assisted diffraction Computational Tomography (SR-FACT). This approach uses ODT to visualize three-dimensional structures in the cell with high volumetric imaging speed, temporal resolution, and a wide temporal range. SIM is used to guide the interpretation and identification of the observed structures. With this technique, the authors identified and studied several interorganelle MCS. Mitochondria were found to form close and stable associations with the nuclear membrane and to interact with LD, late endosomes, and lysosomes. Additionally, the authors described new membrane structures named dark-vacuole bodies that originate from the perinuclear region and interact with several organelles, such as the nuclear membrane and mitochondria, on their way toward the PM [46].

### 2.2. TIRFM

TIRFM is considered a super-resolution technique as it provides axial resolutions of ≈100 nm, below the ≈500 nm axial resolution of the diffraction-limited LSCM. It achieves this by illuminating the sample with light rays that are totally internally reflected at the interface of the cover glass and the sample. This light does not penetrate the sample beyond ≈100 nm, so the acquired images are not contaminated by fluorophores from other out-of-focus planes, as in LSCM (Figure 4) [68,69]. When studying MCS, this technique is usually combined with SIM in an approach referred to as TIRF-SIM, which combines the advantages of both techniques and offers axial and lateral resolutions of ≈100 nm.

Two main limitations exist when using TIRFM for the study of MCS. First, the ≈100 nm axial (and lateral, in the case of TIRF-SIM) resolution limits this technique to large contact site structures and broad morphological studies and cannot be used to extract precise molecular distribution in MCS. Second, because TIRFM acquires images from the cover glass/sample interface, it can only provide information about interactions occurring near the PM and not deep within the cell.

Because of its axial resolution, this approach has been extensively used to characterize Ca^2+^ homeostasis between organelles, centered on the ER-PM MCS’s role in cation store changes. TIRF-SIM studies revealed that GRAMD2a, a protein that tethers at the ER-PM MCS in a PI (4,5)P_2_ lipid-dependent manner, organizes these MCS and pre-marks these regions for STIM1 recruitment [47]. Then, when the Ca^2+^ stored in the ER is depleted, the STIM1 transmembrane proteins in the ER translocate into these regions defined by GRAMD2a, which form punctate spots and tubules that accumulate near the PM [48]. STIM1 accumulation induces a transient MCS that opens channels to mediate Ca^2+^ store replenishment [49,50]. Further TIRF-SIM studies established that after this initial interaction, more stable ring-shaped MCS are formed between the ER and PM by the STIM1 and E-syt1 proteins, further accelerating local ER Ca^2+^ replenishment [51]. TIRFM has also revealed that this Ca^2+^ loaded into the ER is exchanged to lysosomes through IP_3_ receptors in ER-lysosome MCS [52]. ER-PM MCS also regulate lipid homeostasis between ER and the PM. TIRFM studies revealed that the Nir2 protein localizes to these MCS and replenishes PI (4,5)P_2_ from the ER to the PM to compensate for PI (4,5)P_2_ consumption during signaling processes [53].

### 2.3. STED Microscopy

STED microscopy circumvents the diffraction limit by the use of structured light. In a conventional diffraction-limited laser scanning microscope, the sample is scanned by a laser that excites the fluorophores in each spot. The image is built by recording the fluorescence emission in each spot. The minimum size of the excitation spot, limited by the diffraction limit, defines, in part, the spatial resolution of the microscope. STED microscopy increases spatial resolution by reducing the effective size of the emitting spot. It achieves this by exciting a spot with a laser pulse, in the same manner as a conventional laser scanning microscope, followed by a second high-intensity donut-shaped beam that de-excites or depletes fluorophores in the periphery. Only the fluorophores on the center of the donut-shaped beam retain their excited state and emit fluorescence in the detection range, effectively reducing the size of the PSF (Figure 5) [70,71,72]. This technique allows lateral resolutions of up to ≈30 nm, sufficient for studying MCS.

Although it provides high enough spatial resolutions, the use of STED to study MCS has been limited by other factors. First, a reduced number of STED-suitable dyes for two- or multi-color labeling of intracellular targets exists. Most STED-compatible dyes are membrane impermeable and therefore do not allow labeling of cellular organelles in vivo. Second, because of the high power of the depletion beam used in STED, light-induced phototoxicity is an inherent limitation of the technique, to which organelles such as mitochondria and the ER are especially sensitive. Finally, the image acquisition time is longer than in other SRFM techniques, which limits the recording of highly dynamic processes. However, despite these limitations, various research works have managed to visualize, with a spatial resolution below 50 nm, MCS between different organelles.

The early generation of novel photostable STED-suitable and membrane-permeable fluorescent dyes has overcome the first limitation allowing the simultaneous tracking of intracellular organelles and the dynamics of MCS in living cells. The first dual-color live-cell STED image of intracellular organelles was obtained by targeting Halo-tag to the ER and SNAP-tag to the mitochondria for subsequent labeling of these substrates with membrane-permeable reactive dyes SiR-chloroalkane and ATT0590-benzylguanine, respectively. This strategy allowed the visualization of dynamic events occurring at the ER and mitochondria, including apposition between both organelles [54]. The Halo-tag/SNAP-tag strategy has also been used to specifically track ER-mitochondrial dynamics within neurites, where ER tubules wrapping and constraining mitochondria were also imaged [55]. Although it had already been established that mitochondrial pre-constriction by ER tubules is required for mitochondrial fission [7], this work, among others, has demonstrated that every ER-exerted mitochondrial narrowing does not necessarily lead to mitochondrial fission. While in the aforementioned works, ER-mitochondria dynamics were followed by labeling the mitochondrial outer membrane, specific labeling of the mitochondrial inner membrane (MIM) by using PK Mito Orange (a cyclooctatetraene-conjugated Cy3.5) has also been demonstrated to be effective to track ER-mitochondria dynamics in living cells, also allowing the study of MIM architecture and dynamics [56].

Highly dynamic MCS processes are difficult to study with this technique because of its image acquisition time limitation. One strategy to obtain some information about these junctions using STED is by fixing the cells, which implies the loss of dynamic information. This strategy was successfully used to study the implication of ER-PM MCS in autophagosome formation during a stress response [25]. By the triple-labeling of ER (SEC61B-GFP), ER-PM MCS (mCherry-E-Syt3), and autophagosome marker LC3 (immunolabeling), authors could detect the formation of autophagic structures near ER-PM MCS [25]. Human skeletal muscle tissue was also fixed to study PLIN2/PLIN5 LD coat proteins distribution. STED images revealed a preferential localization of PLIN5 in LD-mitochondria MCS [57], which shows that this technique not only serves to study the structure and dynamics of MCS but also the priority localization of proteins within these nanodomains.

Lately, a conceptually related widefield alternative to STED known as Reversible Saturable OpticaL Fluorescent Transition (RESOLFT) microscopy, and especially the recently developed implementation of Molecular Nanoscale Live Imaging with Sectioning Ability (MoNaLISA) has emerged as a suitable tool to reduce both time acquisition and phototoxicity. This approach combines SIM´s patterned light concept with the fluorophore depletion from STED. First, the sample is illuminated with a patterned square (instead of the lineal pattern in SIM) grid-like excitation light of a certain periodicity, so only fluorophores in some areas of the sample are excited. Then, a second patterned square grid-like depletion light is used, with a different periodicity, which depletes some of the excited fluorophores and increases resolutions to sub-diffraction levels. This technique provides 30–65 nm lateral resolution while maintaining a high temporal resolution, enabling the imaging of structures in the entire cell in three dimensions and with relatively short acquisition times [73]. Its implementation has allowed ER-mitochondria organization to be resolved in 2D and 3D in living cells, and through 3D volumetric imaging, authors could visualize an ER tubule wrapping around mitochondria [55].

### 2.4. SMLM

SMLM techniques include PALM, STORM, and PAINT, three related approaches developed almost simultaneously. They are based on the stochastic cycling of fluorescent dyes between the on and off states, so in each frame, only a subset of fluorophores is excited and will emit fluorescence, avoiding overlapping of excitation signals. For each frame, the center-of-mass of the PSF of each fluorophore in the subset is calculated to determine its position, and the data from each frame is compiled to reconstruct a super-resolved image (Figure 6) [74,75,76,77]. These techniques differ primarily in the fluorophores used; PALM uses photoactivatable, photoconvertible, or photoswitchable proteins (e.g., PA-GFP, PA-mCherry, mEOS, mMAPLE, etc.); STORM relies on fluorescent organic dyes (e.g., cyanines, rhodamines, etc.) combined with a specific imaging buffer that enables fluorophore blinking; and PAINT uses external probes that emit fluorescence only when reversibly bound to the molecule or structure of interest (e.g., Nile Red). These techniques provide lateral and axial resolutions of up to ≈20 nm and ≈50 nm, respectively, and because no depletion laser has to be used as in STED, the implementation of multi-color imaging is relatively common.

The main limitation of this technique when studying MCS is that they require specialized fluorophores and certain imaging media for efficient photoactivation or photoswitching. Additionally, these techniques have poor temporal resolution and are not optimal for live cell imaging as they require the acquisition of thousands of frames to build a single super-resolved image [36]. Thus, the majority of studies using these techniques exploit the high spatial resolution they provide while sacrificing the dynamic information of MCS.

STORM and organelle-targeting specialized fluorophores have been used to study interactions between ER and mitochondria. The high spatial resolution (≈30 nm) provided by STORM allowed the observation of tubular extensions connecting neighboring mitochondria during fusion and fission processes, and two-color imaging revealed that the ER is involved in mitochondria constriction and fission through the formation of tubular structures [58]. The combination of STORM with SIM has also revealed that Mitochondrial Rho GTPase (Miro) is heavily involved in the normal formation of ER-mitochondria MCS, and in their absence, interactions between these organelles decrease, leading to alterations in mitochondrial Ca^2+^ uptake and Ca^2+^ concentration in the ER [59]. Using STORM, Mehlitz et al. described the formation of ER-vacuole MCS during *Simkania* bacterial infection. Once the bacterium has entered the host cell, it multiplies within the boundary of vacuoles known as *Simkania*-containing vacuoles (SCV). During the growth of these structures, ER is recruited and interacts via MCS with SCVs. The authors also described the association of mitochondria with these structures, an interplay that starts from an intimate intertwining of mitochondria in premature vacuoles to mitochondrial lining after SCV maturation [78].

Other authors have combined SMLM approaches, such as PALM with TIRF-SIM, to study ER-PM junctions. Using markers specific for ER-PM MCS (tdEos4-MAPPER), PALM allowed an in-depth characterization of the MCS structure, revealing a high morphological heterogeneity. TIRF-SIM confirmed that this heterogeneity is related to the spatial coordination of the ER-PM MCS with the cortical actin cytoskeleton and that ER-PM junction motion is constricted by cortical actin [60]. ER-PM MCS have also been characterized by PAINT, revealing that lipid diffusivity in the ER membrane is reduced in these MCS. This lower diffusion of lipids is not caused by the higher lipid order in the PM, as could be expected, but presumably because of local protein crowding in the ER-PM MCS. Authors postulate that this effect could stabilize the MCS and promote material exchange between the ER and PM [61].

## 3. Concluding Remarks and Outlook

The MCS have gained a special interest in the last decade because of the multitude of cellular functions they perform, which are seriously affected in many diseases such as cancer and neurodegenerative disorders. These observations have revealed that MCS are potential therapeutic targets, but developing new drugs directed to these regions requires previous knowledge of MCS structure, composition, and function. In this regard, the high diversity in MCS—virtually every organelle interacts with each other—and the technical limitations of studying intracellular nanodomains have been an impediment, mainly regarding obtaining dynamic information. SRFM techniques have overcome these limitations allowing visualization of MCS-resident proteins in their specific location, following Ca^2+^ flow between contacting organelles, and even visualizing mitochondrial constriction by ER in living cells.

The SRFM approaches reviewed in this work provide different advantages and have specific limitations. Thus, depending on the research question to address, a different SRFM technique must be chosen. In some cases, several approaches can be combined to produce a more holistic and complete picture of the structure and dynamics of MCS. Some of the summarized techniques provide spatial resolutions on the same scale as the size of the whole MCS. In contrast, others allow us to peer into the distribution of the molecular tethers that constitute them. The temporal resolution of a technique also defines the questions it can answer; approaches with high temporal resolutions allow the study of interorganelle MCS dynamics in living cells, while those techniques with low temporal resolutions are limited to more static structural and morphological analysis. SIM and TIRFM provide lateral and axial resolutions of ≈100 nm, respectively, and can be combined to obtain images with both lateral and axial resolutions in that range. The spatial resolution of these techniques is not enough to precisely define the distribution of the molecular components of MCS, but because they image the sample with widefield illumination, large areas of the cell can be characterized at once. These approaches allow the identification of new interorganelle contacts and provide enough information to describe the general morphology and function of MCS, leading to important discoveries in the field.

SRFM approaches with higher spatial resolution in the range of 20–30 nm, such as STED microscopy and SMLM, enable us to peer into the structure of individual MCS and the distribution of tethers within. Although SMLM uses widefield illumination, it requires the acquisition of several frames to reconstruct a super-resolved image, severely limiting the temporal resolution. Thus, its use is usually limited to fixed samples. Even so, multi-color imaging with SMLM is relatively easy because a wide variety of specialized fluorophores exist. This is of special interest because MCS, by definition, contains molecules of two different organelles. On the other hand, multi-color imaging with STED is more complicated as it requires either the use of several fluorophores that are depleted at the same wavelength or several depletion lasers. Nevertheless, two-color STED has been successfully implemented in the study of MCS, demonstrating its capability in the field. Additionally, contrary to SMLM approaches, 3D imaging with STED microscopy is relatively straightforward, which is of special interest in studying volumetric structures such as MCS.

SRFM is a continuously developing field, with recently developed approaches such as MINimal emission FLUXes (MINFLUX) nanoscopy providing spatial resolutions beyond STED and PALM/STORM/PAINT. MINFLUX combines photoswitchable fluorophores with a donut-shaped patterned excitation light to determine the localization of a molecule with spatial resolutions of up to 2 nm in a few milliseconds [79,80]. MINFLUX also allows the reconstruction of 3D super-resolved multi-color images [81], and although it has not been used to study MCS yet, it has revealed the suborganelle distribution of the mitochondrial MICOS proteins with 3D localization precision of up to 5 nm, demonstrating its application in organelle studies [82]. Although imaging of large areas with MINFLUX still requires long acquisition times and its use to study whole MCS is thus limited, the further development of this and other SRFM techniques and their application in the MCS field could provide increasingly precise information about the molecular components of these junctions.

Most of the SRFM studies mentioned in this review have centered on the proteins populating the MCS. Comparatively, the knowledge about lipid composition, lipid dynamics, and their functional relevance in these membrane junctions is limited. The specific lipid composition and their nanoscale organization in biological membranes play a crucial role in regulating the activity and distribution of many membrane-associated proteins [83,84,85,86,87,88]. In addition to their potential role as protein regulators, lipids such as PI (4,5)P_2_ are also directly involved in several signaling processes [89]. Considering that MCS are lipid bilayer structures that contain many membrane-associated proteins, that some of these junctions are involved in signaling processes, and that they are hubs for the synthesis and transport of some lipids, the lipids populating MCS are expected to fulfill many of the functions observed in other biological membranes.

Although some information on the lipid composition, homeostasis, and their functional relevance on MCS exists [3,13,53,61], very few studies have delved into their spatio-temporal organization, mainly because of the limited amount of suitable probes to study lipid localization and dynamics by SRFM. To overcome this, in recent years, a considerable effort has been made to design, develop, and apply SRFM-compatible probes to study lipids and lipid nanodomains [19,20,90,91], opening a completely new research scenario. Future studies in the field of MCS should make use of these emerging tools to study the lipid components, in addition to proteins, to provide a complete picture of these interorganelle junctions, better understand the regulation of their functions, and potentially discover new therapeutic targets.

In summary, no single SRFM technique can answer all the questions regarding MCS, but each of the different approaches can provide important information, from the dynamics and general morphology of MCS to the spatial distribution of molecules within. In some cases, several of these techniques can be combined, so the information they provide is merged, and a more comprehensive image of the studied MCS arises. To date, most studies have centered on the protein components of MCS, but emerging approaches should allow researchers to additionally investigate the lipids populating these structures, their dynamics, and the role they play in the function of MCS.

## Figures and Tables

**Figure 1 ijms-23-15354-f001:**
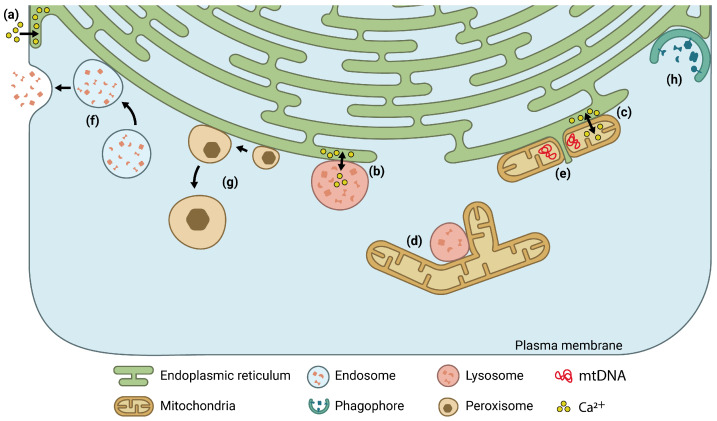
Schematic representation of the main MCS studied and characterized by SRFM. (**a**) ER-PM MCS regulate Ca^2+^ influx from the extracellular environment to the ER in order to replenish its storage; (**b**) ER-stored Ca^2+^ can then be exchanged to the lysosomes and (**c**) mitochondria through the corresponding organelle juxtapositions; (**d**) interactions between mitochondria and lysosomes have also been described; (**e**) contact sites with ER also regulate mitochondrial dynamics and morphology, and couple mtDNA synthesis and distribution with mitochondrial division; (**f**) contacts with ER allows the transport of late endosome to the PM; (**g**) ER-peroxisome droplet MCS regulate peroxisome growth, cholesterol homeostasis, and the synthesis of plasmalogens; (**h**) biogenesis of autophagosome commence at the ER-PM MCS.

**Figure 2 ijms-23-15354-f002:**
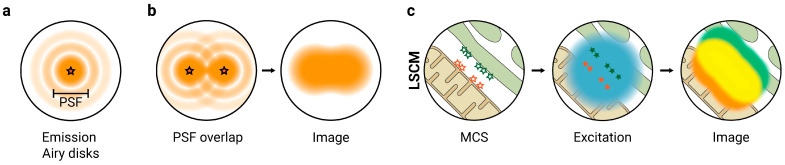
Principles and limitations of optical light microscopy. (**a**) When the light emitted by a fluorophore interacts with the microscope, it is diffracted in a pattern called Airy disks. This diffraction blurs out any light emitted by a single fluorophore to a certain minimal size bigger than the emitter, known as the Point Spread Function (PSF); (**b**) if the distance between two fluorophores is too small, their PSFs overlap and the fluorophores are seen as a single object in the resulting image and cannot be resolved; (**c**) in laser scanning confocal microscopy (LSCM), all excited fluorophores emit light simultaneously, and their PSFs overlap. Thus, LSCM resolution is limited by diffraction, and MCS structures cannot be easily distinguished. Lateral resolution > 200 nm; axial resolution > 500 nm.

**Figure 3 ijms-23-15354-f003:**
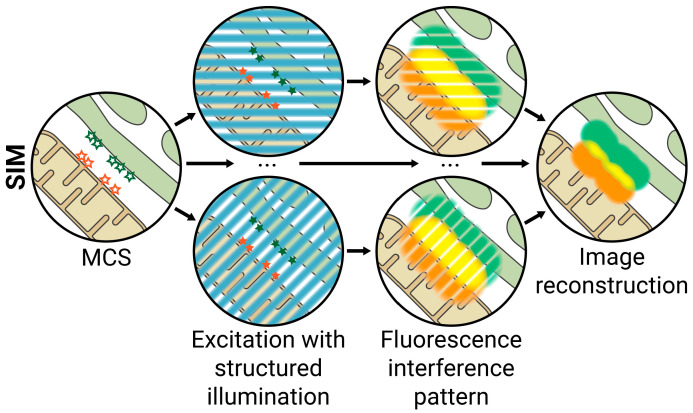
Structured Illumination Microscopy (SIM). The sample is excited by a grid-like patterned light with alternating bands of high and low intensity. This patterned excitation interferes with variations in fluorophore density and creates interference patterns that can be used to extract information about the probe density. The excitation light is rotated, and several interference patterns are combined to reconstruct a super-resolved image. Lateral resolution ≈ 100 nm; axial resolution ≈ 300 nm.

**Figure 4 ijms-23-15354-f004:**
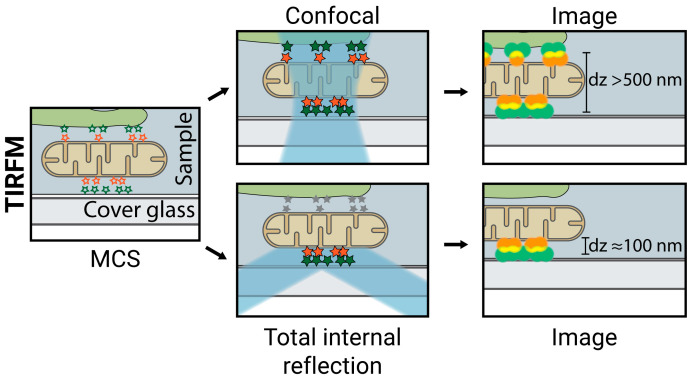
Total Internal Reflection Microscopy (TIRFM). In contrast to LSCM, in TIRFM, the sample is illuminated with light that is totally internally reflected at the interface of the cover glass and the sample and does not penetrate the sample beyond ≈100 nm. The acquired images are not contaminated by fluorophores from other out-of-focus planes. Lateral resolution > 200 nm; axial resolution ≈ 100 nm.

**Figure 5 ijms-23-15354-f005:**
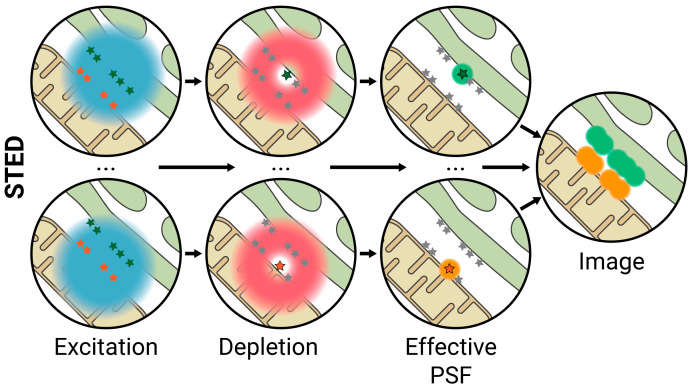
Stimulated Emission Depletion (STED) microscopy. After a given point is excited with a scanning laser, a second high-intensity donut-shaped beam is used to de-excite or deplete fluorophores in the periphery of the spot, so only the fluorophores on the center will emit fluorescence in the detection range, avoiding overlapping of excitation signals. Lateral resolution ≈ 30 nm; axial resolution ≈ 100 nm.

**Figure 6 ijms-23-15354-f006:**
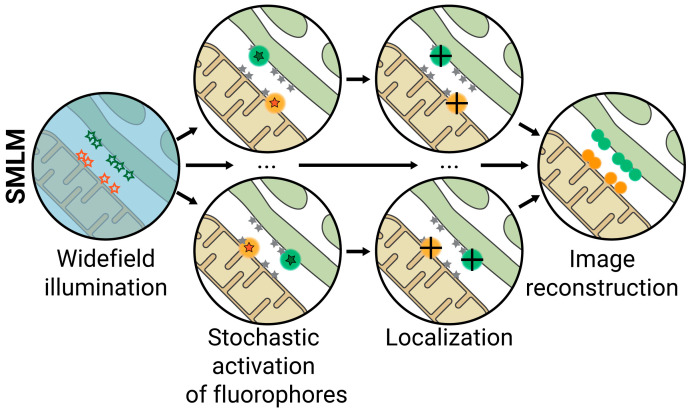
Single Molecule Localization Microscopy (SMLM). Fluorescent dyes that stochastically cycle between the on and off states are used, so in each frame, only a subset of fluorophores is excited and will emit fluorescence, avoiding overlapping excitation signals. For each frame, the center-of-mass of the PSF of each fluorophore is calculated to determine its localization (black cross) and reconstruct a super-resolved image. Lateral resolution ≈ 20 nm; axial resolution ≈ 50 nm.

**Table 1 ijms-23-15354-t001:** Main characteristics, advantages, and limitations of SRFM techniques applied to the study of MCS.

SRFM Technique	Maximum Lateral (d) and Axial (dz) Resolutions (nm)	Advantages	Limitations	References
SIM	d ≈ 100, dz ≈ 300	High sensitivity, common fluorophores, allows 3D imaging	Limited spatial resolution	[13,38,39,40,41,42,43,44,45,46]
TIRFM	d ≈ 200, dz ≈ 100	High axial resolution, common fluorophores	Only images close to PM, limited lateral resolution	[47,48,49,50,51,52,53]
STED	d ≈ 30, dz ≈ 100	High lateral and axial resolutions, allows 3D imaging	Limited multi-color imaging, photostable fluorophores	[25,54,55,56,57]
SMLM	d ≈ 20, dz ≈ 50	Very high lateral and axial resolutions	Specialized fluorophores and buffers, limited temporal resolution, limited 3D imaging	[58,59,60,61]

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
