# Peer review of "Super-Resolution Microscopy to Study Interorganelle Contact Sites"

_ijms, 2022, doi:10.3390/ijms232315354_

Round 1
Reviewer 1 Report
The Review describes the utility of super-resolution microscopy for studying interorganelle contact sites. Scientifically the work is acceptable, but moderate English changes are required.
Author Response
We thank reviewers for their critical comments and important suggestions that helped us to improve our manuscript significantly. Any changes made to the main manuscript file have been introduced with “Track Changes” in the revised manuscript.
Reviewer #1:
The Review describes the utility of super-resolution microscopy for studying interorganelle contact sites. Scientifically the work is acceptable, but moderate English changes are required.
We thank reviewer 1 for his/her comments. We agree that the manuscript could be improved regarding English use and following reviewer 1’s recommendation, the manuscript has been revised and corrected by a native English speaker colleague.
Reviewer 2 Report
The review describes high resolution microscopy techniques which have emerged as a powerful tool for studying interorganelle membrane contact sites (MCS). Here, the main biological functions of MCS and summarize the different approaches for studying them were performed.
In my opinion the review is well written and is a good source of knowledge on the aforementioned subject. However, I have one issue for the authors which should be improved or considered by them.
In my opinion, it should be worth it if the authors described more examples of using such techniques in study on MCS. Description of a few specific researches using super-resolution microscopy will be needed here.
Author Response
We thank reviewers for their critical comments and important suggestions that helped us to improve our manuscript significantly. Any changes made to the main manuscript file have been introduced with “Track Changes” in the revised manuscript.
Reviewer #2:
The review describes high resolution microscopy techniques which have emerged as a powerful tool for studying interorganelle membrane contact sites (MCS). Here, the main biological functions of MCS and summarize the different approaches for studying them were performed.
In my opinion the review is well written and is a good source of knowledge on the aforementioned subject. However, I have one issue for the authors which should be improved or considered by them.
In my opinion, it should be worth it if the authors described more examples of using such techniques in study on MCS. Description of a few specific researches using super-resolution microscopy will be needed here.
We thank reviewer 2 for his/her comments. Regarding the number of referenced works, it must be taken into account that several limitations exist when studying membrane contact sites (MCS): they are highly dynamic and small structures that involve membranes from two different sources. This casuistic restricts the number of available techniques to study them, as they must provide high temporal and spatial resolutions and allow for two- or multi-color imaging. Because of these limitations, the relatively new nature of some of the super-resolution microscopy techniques, and the lack in some cases of suitable fluorescent probes, the utilization of super-resolution techniques in the field has sometimes been limited. Some of these limitations have been overcome recently with different approaches, reflected in an increased number of publications where MCS have been studied by super-resolution imaging in recent years. However, the total number of works is still relatively low, and we believe that our review covers most of the relevant publications.
Nevertheless, we agree that some additional examples could be added to our work, and following reviewer 2’s recommendation, we have performed an additional bibliographical search. We have added to the manuscript several references involving the study of MCS by super-resolution imaging, most of them very recently published (1 from August 2022, 1 from September 2022, 1 from October 2022), as well as a reference from 2014 that we did not add in the original manuscript. Additionally, we have tried to give a more detailed overview of the work carried out in these new references.
We believe that the addition of these new examples has improved the quality and relevance of our review paper, and we thank reviewer 2 again for his/her comments.
Reviewer 3 Report
In this review, Nieto-Garai et al. provide a comprehensive description of the different super-resolution microscopy techniques nowadays available to stydy interorganelle contact sites.
Overall, the work is well-written and well-organized, giving the reader a wide view not only of the potentialities and limitations of different approaches, but also describing the already explored applications.
I have only some minor comments related to the manuscript:
- in the Introduction (lines 56-60), the authors should include in the references papers from the group of Prof. Maulucci (e.g. https://doi.org/10.1016/j.aca.2020.04.076 and https://doi.org/10.3390/ijms22063106) which describe and discuss novel approaches for studying and describing contact sites relying on traditional confocal techniques;
- in Section 2.1, line 186, add a comma: SIM does not require specialized fluorophores or labelling techniques, has high sensitivity...
Author Response
We thank reviewers for their critical comments and important suggestions that helped us to improve our manuscript significantly. Any changes made to the main manuscript file have been introduced with “Track Changes” in the revised manuscript.
Reviewer #3:
In this review, Nieto-Garai et al. provide a comprehensive description of the different super-resolution microscopy techniques nowadays available to study interorganelle contact sites.
Overall, the work is well-written and well-organized, giving the reader a wide view not only of the potentialities and limitations of different approaches, but also describing the already explored applications.
I have only some minor comments related to the manuscript:
- in the Introduction (lines 56-60), the authors should include in the references papers from the group of Prof. Maulucci (e.g. https://doi.org/10.1016/j.aca.2020.04.076 and https://doi.org/10.3390/ijms22063106) which describe and discuss novel approaches for studying and describing contact sites relying on traditional confocal techniques;
- in Section 2.1, line 186, add a comma: SIM does not require specialized fluorophores or labelling techniques, has high sensitivity...
We thank reviewer 3 for his/her comments. We have introduced the proposed corrections in section 2.1. In relation to the work by the group of Prof. Maulucci, we highly appreciate the recommendation from reviewer 3, as their work comprises an important and elegant approach to study the composition and distribution of specific lipids in living cells. We have added the first reference https://doi.org/10.1016/j.aca.2020.04.076 in the introduction as suggested by reviewer 3. Regarding the second reference https://doi.org/10.3390/ijms22063106, the authors designed polarity sensitive fatty acid derivatives and studied their distribution in organelles, but they do not specifically center on MCS. Since the tools they describe could be very valuable in order to study lipid organization in MCS, we have added a mention to this second reference in section 3, “Concluding remarks and outlook”, line 537, when new tools to study lipid distribution and dynamics are mentioned.